# Uni-Match: A Semantic Unified Model for Query-Product Retrieval

**Zhenyang Zhu[1], Rui-jie Zhu[2], Yunrui Ge[3], Qihang Zhao[1],***

[1]Kuaishou Technology Co. Ltd, Beijing, China
[2]University of Electronic Science and Technology of China, Chengdu, China
[3]Wuhan University of Technology, Wuhan, China
`zhuzhenyang@kuaishou.com`, `ridger@std.uestc.edu.cn`,
`2282419633@qq.com`, `zhaoqihang@kuaishou.com`

## Abstract

For most practical search systems, the cascaded matching-prerank-rank architecture is designed. In the prerank stage, the dual-tower structure is widely used to maintain efficiency. However, due to the lack of interaction between query and document, this architecture could only take into account efficiency but not both effectiveness and efficiency. Inspired by this, we propose a simple but effective dual-tower model: **uni-match**, which has the efficiency of dual-tower model and the effectiveness close to that of cross model. Sufficient offline and online experiments show the effectiveness of our proposed method. Currently, the Uni-match model has been deployed in the search system of a shortvideo App, providing daily services to hundreds of millions users.

## 1 Introduction

The prerank module between the matching and ranking modules in the search systems is required to acquire the efficiency of matching and the effectiveness of ranking. Due to the performance limitation of the prerank module, the current research focus is on how to achieve the effectiveness of the cross model under the efficiency of the dual-tower model. The general approach is to conduct Q(uery)-D(ocument) interaction at the top of the dual-tower model, such as token-level (ColBERT Khattab and Zaharia (2020)) and sentence-level (DPR Karpukhin et al. (2020)). Nonetheless, these methods do not take into account the semantic information of both token and sentence-level. Inspired by this, we propose a simple but effective interaction mechanism: Union-interaction, which maps the query and document sentences and token-level information to the unified semantic space for matching. Meanwhile, in order to learn complex semantic information, we adopt the easy-to-difficult age curriculum learning training strategy. Our model achieves significant improvements in semantic matching performance, leading to increased customer satisfaction and potentially higher revenue for businesses.

## 2 Method

### 2.1 Model Architecture

Fig. 1 depicts the architecture of model, which comprises: (a) a query encoder $f_Q$, (b) a document encoder $f_D$, and (c) the union-interaction mechanism. For query encoder $f_Q$ and document encoder $f_D$, we apply 6-layer BERT Devlin et al. (2018) encoder to encode raw text. Then we adopt the union-interaction mechanism to learn the token-level and sentence-level semantics between query and document. The details will be introduced in Section 2.2.

### 2.2 Union-interaction

---

*Corresponding Author.

In the prerank stage, different from the interaction-based model commonly used in the ranking stage, the mainstream semantic matching model is the dual-tower model, which is able to encode documents into embeddings offline and extract the embeddings directly online for efficiency. Hence, the architecture is designed as a dual-tower structure to reduce the interaction between query $q$ and document $d$, while for this reason, it has certain defects in effectiveness. To alleviate the problem of incompatibility between efficiency and effectiveness, we propose an union-interaction mechanism. Specifically, given the representation of $q$ and the $d$, we use the vector corresponding to '[cls]' of the query encoder as the query, perform the attention Vaswani et al. (2017); Luong et al. (2015); Bahdanau et al. (2014) operation on the representation of the $q$ and the $d$, and unify the representation of the query and the representation of the $d$ to a semantic space for matching:

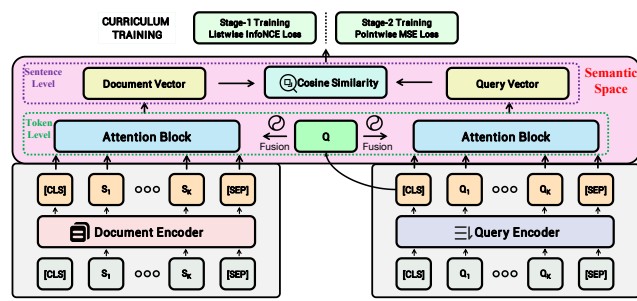

Figure 1: The framework of Uni-match.

$$
\begin{aligned}
\mathcal{V}^q &= softmax(\frac{Q_q K_q^T}{\sqrt{d}})V_q, \quad Q_q = E_{q_{cls}}W_Q, \quad K_q = E_q W_K^q, \quad V_q = E_q W_V^q \\
\mathcal{V}^d &= softmax(\frac{Q_q K_d^T}{\sqrt{d}})V_d, \quad Q_q = E_{q_{cls}}W_Q, \quad K_d = E_d W_K^d, \quad V_d = E_d W_V^d
\end{aligned}
\tag{1}
$$

Where $\{W_Q, W_K^q, W_V^q, W_K^d, W_V^d\} \in \mathbb{R}^{D \times d}$ are learnable parameters, $\{E_q, E_v\} \in \mathbb{R}^{N \times D}$ are the representation of $q$ and $d$, and $\{\mathcal{V}^q, \mathcal{V}^d\} \in \mathbb{R}^{N \times d}$ are the representation of $q$ and $d$ unified into the same semantic space, and learn token and sentence-level semantic matching information of $q$ and $d$ simultaneously, as shown in Fig. 1.

## 2.3 CURRICULUM TRAINING

In order to better learn the extremely complex semantic matching information between query and document, we adopt a two-stage curriculum training strategy from easy to difficult. In the 1st stage, based on the list-wise training paradigm, we use the infoNCE Oord et al. (2018) loss to force the model to learn the rank ability of the ranking model for documents. In the 2nd stage, based on the point-wise training paradigm, we use MSE loss to learn the scoring ability of the fine-tuning model.

## 3 EXPERIMENTS

Table 1: Online A/B test results.

| Model | GMV | Volume of Order | User Order Completion |
|---|---|---|---|
| Base | +0% | +0% | +0% |
| **Uni-match** | **+5.826%** | **+2.261%** | **+1.881%** |

We evaluate the proposed method on the offline kuai_search, MS MARCO dataset (illustrated in Tab. 2 and Tab. 3 in Appendix) and conducted an online A/B test for 30 days to verify the effectiveness and efficiency of Uni-match (shown in Tab. 1, Other details are in the Appendix). In terms of AUC metric, our Uni-match surpasses other semi-interaction-based models by 2.69pp, doing nearly as well as certain interaction-based models, which need more computational resources (As shown in Tab. 2, for a single $q - d$ pair, the calculation time of the interaction-based model 'cross match' is 127 ms, while uni-match is only 22 ms.). In online A/B testing, we could observe that our model had significantly improved the system overall based on the three important metrics of Gross Merchandise Volume (GMV), Volume of Order, and User Order Completion.

## 4 CONCLUSION

In this paper, we propose a novel interaction mechanism: union-interaction, which can unify the representation of query and document into a semantic space for matching. Meanwhile, we aggregate the token significance of the query and the document into the sentence-level semantics of the query and the document using the attention mechanism. The results of sufficient experiments support the effectiveness of our proposed method.

## 5 URM Statement

The authors acknowledge that all authors of this work meets the URM criteria of ICLR 2023 Tiny Papers Track.

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

## Appendix

### Datasets

**kuai_search Dataset.** We constructed a dataset containing 120,000 samples through the scoring results of the refined ranking model, combined with manual annotation, in which positive samples: negative samples = 51%: 49%.

### Baselines

**Cross Match.** Interaction-based semantic matching model (cross model).

**ColBERT Khattab and Zaharia (2020).** Representation-based model (Dual-tower model), a weak interaction mechanism of late-interaction is introduced at the top of the model (token-level).

**DPR Karpukhin et al. (2020).** Representation-based model, use dot-product at the top level of the model for sentence-level interaction.

**Poly-encoder Humeau et al. (2020).** Representation-based model, a learnable one-way interaction mechanism is introduced at the top of the model.

PARAMETER SETTINGS.

We give some key parameter configurations: the size of hidden layer $D$ is 768, the number of hidden layer is 6, the number of attention heads is 12, the size of $\mathcal{V}^d$ and $\mathcal{V}^q$ are 24, and learning rate $lr = 0.0002$.

OFFLINE RESULTS.

Table 2: Evaluation results of uni-match and baseline methods on the kuai_search dataset.

| Type | Model | AUC | PNR | Time (ms) |
|------|-------|-----|-----|-----------|
| Interaction-based Model | Cross Match | **72.78** | **1.96** | 127 |
| Dual-tower Model | ColBERT Khattab and Zaharia (2020) (6-Layers) | 63.28 | 1.57 | 38 |
| | ColBERT (12-Layers) | 66.56 | 1.61 | 63 |
| | DPR (Karpukhin et al. (2020)) | 67.23 | 1.65 | 15 |
| | Poly Encoder Humeau et al. (2020) | 70.05 | 1.68 | 29 |
| | **Uni-match (6-Layers) (Ours)** | **72.74** | **1.72** | 22 |

Table 3: Evaluation results of uni-match and baseline methods on the MS MARCO dataset.

| Type | Model | AUC | PNR | Time (ms) |
|------|-------|-----|-----|-----------|
| Interaction-based Model | Cross Match | **83.61** | **2.28** | 89 |
| Dual-tower Model | ColBERT Khattab and Zaharia (2020) (6-Layers) | 70.53 | 1.77 | 29 |
| | ColBERT (12-Layers) | 73.41 | 1.92 | 48 |
| | DPR (Karpukhin et al. (2020)) | 75.02 | 1.96 | 12 |
| | Poly Encoder Humeau et al. (2020) | 78.81 | 2.06 | 21 |
| | **Uni-match (6-Layers) (Ours)** | **81.17** | **2.11** | 15 |

