# OpenReview forum: "Uni-Match: A Semantic Unified Model for Query-Product Retrieval"
_ICLR.cc/2023/TinyPapers — Submitted to Tiny Papers @ ICLR 2023_

### Official Review · Reviewer_5P3y · 2023-03-24

**Confidence:** 4

**Summary Of Contributions:**

The paper adds a union-interaction mechanism to the twin-tower model for increased matching effectiveness. The proposed uni-match model is evaluated in both offline and online settings and the performance increase against baseline models is significant.

**Rating:**

High Potential (HP): a submission which meets the reviewing criteria and has potential to make an impact on the field

**Strengths And Weaknesses:**

Strengths:
1. The paper is well-written in terms of explaining research motivations and diversifying itself from other twin-tower models with query document interactions such as ColBERT and DPR.
2. The performance improvement is significant when compared to other twin-tower models such as ColBERT and DPR.
3. The model is more efficient when compared to interaction-based model.

Weaknesses:
1. For offline experiment, the work is only tested on private offline dataset (kuai_search dataset) which makes it impossible to reproduce or validate the correctness of the reported performance. Same goes for online testing.
2. Minor issue but the format seems off in several places, such as Figure 1 and Table 1.

**Suggested Changes:**

1. Replace offline experiments with a public dataset or make kuai_search dataset public.

---

### Official Review · Reviewer_dVH8 · 2023-04-02

**Confidence:** 3

**Summary Of Contributions:**

The paper proposes a dual tower model based on an interaction mechanism which maps the query and document sentences and token-level information to the unified semantic space for matching, called Uni-Match. The paper further shows the effectiveness of the proposed approach via offline and online experiments

**Rating:**

High Potential (HP): a submission which meets the reviewing criteria and has potential to make an impact on the field

**Strengths And Weaknesses:**

Strengths
* Clarity : Establishes the research problem clearly and experimental design is clearly defined to support the better performance of proposed model
* Correctness : The model description is sound and the background has been established from previous literature. The multiple metrics used to evaluate the model are sensible and relevant showing improvement across both offline and online experiments which is essential for e-commerce search
* Reproducible : No code is provided however, the training strategy and parameter settings are defined clearly. The Union-interaction strategy used in the paper is defined mathematically. Offline Experiments are on a private dataset.
* Follows basic requirements : Yes, page limit and URM followed

Weaknesses
1. The intuition as to why the union-interaction would be beneficial is not established properly
2. Offline experiments are not reproducible as the dataset is not open source






**Suggested Changes:**

The paper in itself is complete and provides sufficient evidence to support the improvements of the proposed model. I would suggest to add some intuition in simple language before section 2.2 to introduce Union-interaction and why it should improve the performance

Minor formatting errors on Page 2, Figure 1 and Table 1

---

### Author Response · Authors · 2023-05-30
**Revise feedback and Acknowledgement**

We sincerely thank the reviewers for their constructive comments on our paper. We have made corresponding improvements in the revised version, including adding comparative experiments with public benchmark.

---

### Author Response · Authors · 2023-05-30
**Opt-in for Archival**

We would like to opt-in for archival. Thank you!

---

### Meta-Review · Area_Chair_Q56t · 2023-04-02

**Recommendation:** Invite to present
**Confidence:** 4

**Metareview:**

Pros
- Overall the paper is well-written and the results seem sound, well-motivated and of relevance to the community.
- The proposed model is clearly described, well-distinguished from related literature and seems to attain useful combined effectiveness and efficiency goals as claimed.

Cons
- Code and certain data set is private.

**Summary:**

The authors propose a dual-tower approach to map the query and document sentences and token-level information to a unified semantic space for matching, and is based on an interaction mechanism. They further show the efficiency and effectiveness of the proposed method via (offline and online) experiments. The model seems sound but some experiments are not reproducible.

**Comments And Feedback To The Authors:**

Please make a careful pass to fix formatting issues including those indicated by the reviewers, e.g.
- "architec- ture" and "exper- iments" in openreview abstract
- Figure 1, Table 1.

Highly recommended:
- Consider having a public version of your code linked in from the paper, and ensure that the experiments are run on public data sets (e.g. the "kuai_search" data set could be made public if possible, or experiments repeated on alternative public data sets) as it is essential to reproducibility and good science.

**Reason For Not Giving A Higher Recommendation:**

The lack of code and use of private data set is the only challenge to reproducibility, and otherwise the paper could be considered as "notable".

**Reason For Not Giving A Lower Recommendation:**

The recommendation matches assessment of both reviewers.

---

### Decision · Program_Chairs · 2023-04-10

Invite to present